# *Trichoderma asperellum* 22043: Inoculation Promotes Salt Tolerance of Tomato Seedlings Through Activating the Antioxidant System and Regulating Stress-Resistant Genes

**DOI:** 10.3390/jof11040253

**Published:** 2025-03-26

**Authors:** Guangyan Hu, Zhongjuan Zhao, Yanli Wei, Jindong Hu, Yi Zhou, Jishun Li, Hetong Yang

**Affiliations:** 1Shandong Province Key Laboratory of Applied Microbiology, Ecology Institute of Shandong Academy of Sciences, Qilu University of Technology, Jinan 250103, China; hgyyffs2025@163.com (G.H.); zzjfrances@aliyun.com (Z.Z.); yanli_wei@163.com (Y.W.); hujd@sdas.org (J.H.); 2China–Australia Joint Laboratory for Soil Ecological Health and Remediation, Ecology Institute of Shandong Academy of Sciences, Qilu University of Technology, Jinan 250103, China; 3School of Agriculture, Food and Wine, The University of Adelaide, Urrbrae 5064, Australia; yi.zhou@adelaide.edu.au

**Keywords:** NaCl, *Trichoderma asperellum*, seedling growth, scavenging enzymes, gene expression

## Abstract

Salt stress poses a major threat to plant growth, and breeding for salt-tolerant varieties is not always successful to ameliorate this threat. In the present experiment, the effect of *T. asperellum* 22043 inoculation on the growth of salt-stressed tomatoes and the mechanisms by which it improves salt tolerance were investigated. It was observed that tomato plants treated with *T. asperellum* 22043 spore suspension under salt tress (50 and 100 mM NaCl) consistently exhibited higher seeds germination, seedling survival rate, plant height, and chlorophyll content, but lower malondialdehyde and proline contents than the plants treated without the *Trichoderma*. *T. asperellum* 22043 effectively improved the stress resistance of tomato through regulating the transcriptional levels of reactive oxygen species (ROS) scavenging enzyme gene expression to modulate the activity of ROS scavenging enzymes and the expression of the genes related to transporter and aquaporin to maintain the balance of cell Na^+^. In conclusion, *T. asperellum* 22043 can enhance tomato seedlings’ salt tolerance by activating the antioxidant system and regulating the expression of stress-resistant genes.

## 1. Introduction

Soil salinization is one of the major environmental problems affecting terrestrial ecosystems and sustainable agricultural development [1]. It was reported that there are about 930 million hectares of arable land worldwide that are affected by salinization, accounting for more than 6% of the total land area, and this figure continues to grow [2]. In China, saline land is widely distributed, of which the Yellow River Delta area has become the most concentrated area of coastal saline land due to natural and man-made factors [3]. Tomato is one of the most important fruits and vegetables in the world, with high economic and nutritional value, and can be grown in coastal saline lands, not only to improve the utilization of saline soil, but also to bring economic benefits to local development [4]. However, the high salt content of coastal saline soils has led to low survival and yield of tomatoes, limiting agricultural production [5]. Therefore, solving the problem of plant growth in saline soils and alleviating the harm of salt stress on plant growth is the first step to effectively carry out saline land improvement and integrate utilization.

Salt stress not only affects the physicochemical properties of the soil [6], but also limits the growth and germination of plant seeds [7], causes ion imbalance in the plant [8], and leads to a decrease in the photosynthetic rate of the plant [9]. To improve the salt resistance of plants, traditional breeding and transgenic techniques have been used to breed salt-tolerant varieties, but these two approaches are usually costly and time-consuming [10]. Previous works suggested that salt-tolerant microorganisms were effective in promoting salt-stressed plants’ growth and enhancing plant salt tolerance through their interactions with plants [11]. Furthermore, this method has the advantages of low cost and quick effect [12].

*Trichoderma* is a common fungus in nature with strong survival and adaptation ability [13]. It has been proved that *Trichoderma* can promote the growth of plants in saline soil and enhance the resistance of plants to salt stress by colonizing the roots of plants [14]. *Trichoderma* improves the salt tolerance of plants through various regulatory mechanisms. First, *Trichoderma* can increase the uptake of nutrients by plants, such as P, K, Ca, Mg, and Mn [15]. Secondly, *Trichoderma* can produce xylanase, a substance that activates the reactive oxygen species scavenging system and induces plants to increase antioxidant enzyme activity as well as upregulate genes of reactive oxygen species scavenging-related enzymes, thus improving plant resistance to stress [16]. Osmoregulation is an important defense mechanism for plant adaptation to high salt environments. Among them, ion transport proteins in the cell membrane regulate the ion concentration inside and outside the cell and maintain intracellular osmotic pressure [17]. Aquaporins (AQPs) are membrane channel proteins that are responsible for water transport in plants. It is very important to maintain water balance in plants as this can participate in salt stress in plants [18]. What is more, by reducing the osmotic potential of plant leaf cells under salt stress conditions, *Trichoderma* facilitates the maintenance of a high relative water content in plant leaves, which in turn ensures the normal performance of many functions such as photosynthesis [19]. *Trichoderma* can also produce plant growth regulators like α-naphthaleneacetic acid (NAA), indole acetic acid (IAA), and indole-3-butyric acid (IBA) that may alleviate salt stress [20]. Recent studies have found that some *Trichoderma* produce ACC-deaminase to prevent excessive increases in the synthesis of ethylene under salt stress conditions, which is one of the efficient mechanisms to enhance plant tolerance to salt stress [21,22]. And *Trichoderma* can regulate the expression of salt-tolerance-related genes in plants, thus activating their defense ability [23].

Our previous works showed that *T. asperellum* 22043 had salt tolerance, a pro-growth function, and an antagonistic effect against pathogenic fungi, yet the ability of *T. asperellum* 22043 in promoting plant growth and enhancing salt tolerance in plants under salt stress has not been studied. The process by which *Trichoderma* induces plants to develop resistance to salt stress and adapt to salt-stressed environments is a complex endeavor that is the result of the coordination of various mechanisms. In this study, the physio-biochemical characteristics of tomato, the expression of tomato stress resistance genes was determined. The aims were to evaluate the role of *T. asperellum* 22043 in promoting tomato growth under salt stress, explore the possible mechanism of enhancing tomato tolerance to salt stress by *T. asperellum* 22043, and provide a new solution to the problem of saline plant growth.

## 2. Materials and Methods

### 2.1. Trichoderma Strain and Plant Material

The salt-tolerant *T. asperellum* 22043 was isolated from saline soils of Dongting Lake wetlands and kept in the *Trichoderma* spp. repository in the Shandong Province Key Laboratory of Applied Microbiology. This strain was a multifunctional salt-tolerant *Trichoderma* strain that was not only tolerant to biotic as well as abiotic stresses, but also capable of producing IAA. The morphology and DNA sequence (*ITS* and *TEF1* genes) have been published in our previous study [24]. The tomato variety was German tomato MM-S2011, which had the characteristics to adapt to various adversities.

### 2.2. Plate Culture Experiment

Uniformly sized and full tomato seeds were first disinfected with 70% ethanol for 30 s, rinsed three times with sterile water, then disinfected with 3% (*v*/*v*) NaClO for 3 min, and then rinsed five times with sterile water. The disinfected tomato seeds were immersed in a suspension of *T. asperellum* 22043 spore at concentrations of 0, 2 × 10^4^, 2 × 10^5^, and 2 × 10^6^ conidia/mL for 3 h, respectively. Preparation of *Trichoderma* spore suspension was conducted with reference to the method of Zhang et al. [25]. The experimental design was 3 *Trichoderma* spore concentrations × 3 NaCl concentrations × 3 replicates.

Three layers of filter paper soaked with different concentrations (0, 50, and 100 mM) of NaCl solution were laid flat in 9 cm-diameter Petri dishes. Tomato seeds (30 seeds) treated with different concentrations of *Trichoderma* spore suspension or sterile water were placed evenly in each Petri dish and then covered with a layer of filter paper soaked with different concentrations of NaCl solution above the seeds. Petri dishes were cultivated in a light incubator at 25 ± 3 °C for 6 days under 16/8 h day/night cycle, and the tomato seeds’ germination was observed and recorded daily for 6 days. Average shoot length and maximum root length were measured on day 6. Three replicates of each treatment were used.

### 2.3. Pot Experiment

The experimental design was 2 *Trichoderma* treatments (+ and −) × t salinity levels (+ and −) × 3 replicates. Tomato seedlings of equal length and with three or four true leaves were divided equally into two groups. One group soaked the roots with 1L of *Trichoderma* spore suspension at a concentration of 2 × 10^5^ conidia/mL for 30 min, and the other group soaked the roots with 1L of sterile water for 30 min. After root soaking, tomato seedlings of the same treatment were placed in pots containing approximately 250 g of sieved sterile soil, and then 60 g of the same soil was taken to cover the seedlings roots. Each treatment was watered with 100 mL of sterile water, once every two days. On the 7th day after planting, salt stress treatment started with 100 mL of 200 mM NaCl solution irrigated in the NaCl treatment group and 100 mL of sterile water poured in the control group, once every 2–3 days, and repeated three times. Final soil salt content is approximately 1.13%. All plants were grown at 25 ± 2 °C in a greenhouse with 16/8 h day/night cycle conditions. This experiment was conducted in a completely randomized design with three replicates. The seedlings’ survival rate, plant height, and all physio-biochemical indices were determined on the 14th day after treatment of tomato seedlings with NaCl.

### 2.4. Physiological and Biochemical Characteristics in Tomato Seedlings

The leaves of the same parts of tomato seedlings were used to determine various physiological and biochemical parameters. Chlorophyll content was measured with reference to the procedure of Arnon [26]. The methods of Lv et al. [27] was carried out to estimate the malondialdehyde (MDA) content. Proline content in tomato seedlings was determined according to the description by Bates et al. [28]. The Folin–Phenol method was followed to determine the soluble protein content. Tomato leaves (0.3 g) were ground into a homogenate with 3 mL of phosphate buffer (0.1 M, pH 7.5). The homogenate was centrifuged at 12,000 rpm for 10 min. To 1 mL of supernatant was added 5 mL of mixed reagent (Reagent A: dissolve 1 g Na_2_CO_3_ in 50 mL 0.2 M NaOH; Reagent B: 0.5 g CuSO_4_·5H_2_O dissolved in 100 mL 1% sodium potassium tartrate solution, then mix 50 mL A reagent with 1 mL B reagent). After 10 min, 0.5 mL of Folin phenol reagent was added and the absorbance was measured at 500 nm. Bovine serum protein standard curves were calculated.

Antioxidant enzyme activities were determined by spectrophotometer methods. The enzyme solutions were prepared using a known FW of tomato seedlings by homogenizing the sample with 5 mL of 50 mM phosphate buffer (pH 7.8). Superoxide dismutase (SOD) activity was assessed according to the method of Shafi et al. [29]. The peroxidase (POD) activity was observed by the procedure as described by Lurie et al. [30]. For the determination of catalase (CAT) activity, the method of Miyagawa et al. [31] was used. Antioxidant enzyme activities were utilized to evaluate salt resistance of tomatoes.

### 2.5. Expression of Genes Related to Salt-Tolerance

Tomato seedlings leaves and roots in all treatment were used to extract the total RNA following the manufacturer’s recommendations of E.Z.N.A.^®^Plant RNA Kit (OMEGA, Norcross, GA, USA). The first strand cDNA was synthesized using M-MLV reverse transcriptase (TransGene, Beijing, China) according to the manufacturer’s instructions. The reverse-transcribed cDNA products were used as templates for real-time quantitative PCR. The forward and reverse primers for qRT-PCR analysis were designed and synthesized by searching DNA sequences from the Gene bank (Table 1). In each 20 μL reaction system, 10 μL of 2 × SYBA Green Buffer, 1 μL of upstream primer, 1 μL of downstream primer, 20 ng of cDNA template, and 6 μL of deionized water were added. Three biological replicates were performed for each treatment during the quantification. Tomato actin gene was used as an internal standard. The relative expression of ion transporter genes (*HKT*, *SOS1*, *NHX1*, and *LHA4*), oxidative stress genes (*SOD1* and *APX2*), and aquaporin genes (*PIP2-9*) were calculated with the method of 2^−∆∆Ct^ [32] to evaluate the salt tolerance of tomatoes. The PCR reaction procedure comprised 3 min at 95 °C, followed by 45 cycles consisting of 10 s at 95 °C, 30 s at 52 °C, and 30 s at 72 °C. At the end of the cycle, 65 °C was held for 5 s and then increased by 0.5 °C every 5 s to 95 °C to establish the melting curve.

### 2.6. Statistical Analysis

The data were analyzed using SPSS 19.0 (IBM Corporation, Armonk, NY, USA) and compared by one-way ANOVA. Multiple comparisons were performed using Duncan’s new multiple polarization method to test the significance level of differences between treatments (*p* < 0.05). The figures were drawn using OriginPro 2024 (OriginLab Corporation, Northampton, MA, USA).

## 3. Results

### 3.1. Effect of T. asperellum 22043 on Tomato Seed Germination and Seedling Growth In Vitro

Compared with tomato seeds treated without *T. asperellum* 22043, the germination rate of tomato seeds treated with *T. asperellum* 22043 was significantly higher at 0 mM (Figure 1a), 50 mM (Figure 1b), and 100 mM (Figure 1c) NaCl treatment. On the 6th day of seed germination experiment, the highest germination rate of tomato seeds treated with 2 × 10^5^ conidia/mL concentration of *T. asperellum* 22043 spore suspension was 97.8% and 34.4% at 50 mM and 100 mM NaCl, respectively. The optimum *Trichoderma* spore suspension concentration to induce germination of salt-treated tomato seeds was 2 × 10^5^ conidia/mL.

The *T. asperellum* 22043 treatment increased the shoot length (Figure 1d) and root length (Figure 1e) of tomato seeds significantly whether the NaCl concentration was at 50 mM or 100 mM. The seeds treated with 2 × 10^5^ conidia/mL concentration of *T. asperellum* 22043 spore suspension increased the average shoot length by 35% to 50% and the maximum root length by 14% to 32% across the NaCl treatments (50 mM, 100 mM). *T. asperellum* 22043 increased the average shoot lengths at 50 mM NaCl compared to nonsaline control seeds.

### 3.2. Effect of T. asperellum 22043 on Tomato Seedling Growth in Greenhouse

The growth conditions of tomato seedlings treated with *T. asperellum* 22043 under salt stress were shown in Figure 2a. Tomato seedlings treated with *T. asperellum* 22043 improved in height (Figure 2c) by 2% in comparison to control seedlings, respectively. NaCl treatment significantly reduced the growth parameters of tomato plants. However, with *T. asperellum* 22043 application, NaCl-treated tomato seedlings improved in survival rate (Figure 2b) by 76% and seedling height by 5% compared to the NaCl-treated seedlings alone. *T. asperellum* 22043 was able to promote the growth of salt-treated tomato seedlings.

### 3.3. Chlorophyll, MDA, Proline and Soluble Protein Content in Tomato Seedling

The MDA (Figure 3b) and proline contents (Figure 3c) in tomato seedlings increased significantly with the NaCl treatment. The content of chlorophyll (Figure 3a) was increased by 42%, MDA content decreased by 5%, proline content increased by 465%, and soluble protein content (Figure 3d) increased by 11% with the *T. asperellum* 22043 compared with control tomato seedlings. Tomato seedlings with *T. asperellum* 22043 under salt stress increased chlorophyll content by 35%, decreased MDA content by 9%, increased proline content by 72%, and increased soluble protein content by 12% compared to the salt-treated seedlings alone. *T. asperellum* 22043 showed significant effectiveness in enhancing the adaptation of tomato to salt stress by increasing the content of osmoregulatory substances.

### 3.4. SOD, POD and CAT Activity Assay

Compared with the control, the addition of *T. asperellum* 22043 increased SOD activity (Figure 3e) by 14%, decreased POD activity (Figure 3f) by 40%, and decreased CAT activity (Figure 3g) by 53%. The POD activity in tomato seedlings increased significantly with the NaCl treatment. The SOD activity, POD activity, and CAT activity of tomato seedlings treated with *T. asperellum* 22043 under NaCl stress increased by 10%, 10%, and 93% in comparison to NaCl-stressed seedlings alone, respectively. The ability of tomato to scavenge ROS was enhanced. CAT activity in tomato was significantly higher. *T. asperellum* 22043 mainly enhanced tomato resistance to salt stress by increasing the CAT activity.

### 3.5. Gene Expression

*T. asperellum* 22043 induced lower *HKT* (Figure 4a,b) gene expression in salt-stressed tomato roots compared to NaCl treatment alone. It also induced upregulation of *HKT* gene expression in tomato seedlings under NaCl stress.

*T. asperellum* 22043 application induced higher expression of *NHX1* (Figure 4c,d) gene in NaCl treated tomato seedlings. Compared with the NaCl control, addition of *T. asperellum* 22043 induced the decreases of transcriptional level of *SOS1* (Figure 4e,f) and *LHA4* (Figure 4g,h) gene in seedlings, but all were significantly higher than the nonsaline control group.

The expression of *SOD1* (Figure 4i,j) and *APX2* (Figure 4k,l) gene in tomato seedlings and roots were increased under NaCl stress compared to control tomato seedlings and roots. *T. asperellum* 22043 induced upregulation of *SOD1* gene in control tomato seedlings and roots. In addition, the expression of *APX2* gene in roots were also upregulated in *T. asperellum* 22043-treated tomato roots. At the NaCl treatment, the application of *T. asperellum* 22043 led to the lower expression of *SOD1* and *APX2* gene in tomato seedlings and roots compared to NaCl treatment alone.

*T. asperellum* 22043 was able to increase the expression of *PIP2-9* (Figure 4m,n) gene in salt-treated tomato seedlings, which helped tomato to absorb water, while decreasing the expression of *PIP2-9* gene in the roots and preventing excess water from entering the plant.

## 4. Discussion

Salt stress affected the growth and development of plants at all stages and may even cause plant death at higher intensities of salt stress [33]. It was found that plant salt tolerance was not only regulated by functional genes related to salt resistance in plants, but also by the external biological environment, and some microorganisms were shown to be closely related to improving abiotic stress tolerance in plants [34]. Some studies have shown that *Trichoderma* can induce resistance to abiotic stress in plants by establishing interaction with the host plant [35]. In this study, we demonstrated the dual effect of *T. asperellum* 22043 in improving tomato biomass and salt tolerance under salt stress conditions using seed germination as well as pot experiments. On the one hand, *T. asperellum* 22043 effectively improved the stress resistance of tomato through regulating the transcriptional levels of ROS scavenging enzyme gene expression to modulate the activity of ROS scavenging enzymes; on the other hand, *T. asperellum* 22043 adapted the tomato cells to NaCl stress at an overall level through regulating the transcriptional level of the expression of the ion transport genes as well as the aquaporin genes to maintain the balance of Na^+^ as well as water in tomato cells.

Salt stress can reduce the biomass as well as the total chlorophyll content of plants [36]. In the present study, *T. asperellum* 22043 was able to increase the growth of tomato seeds and seedlings treated with salt stress, alleviating the adverse effects of salt stress on plants and maintaining proper plant growth under salt stress conditions [37]. Similarly, *T. reesei* was reported to increase the chlorophyll content of wheat roots under salt stress conditions [38].

Under salt stress conditions, a large amount of ROS was produced in plants, which caused membrane lipid peroxidation damage to plants and produced membrane lipid peroxidation products such as MDA [39], and MDA content in plants can be used to evaluate the degree of exposure to salt stress. During long-term evolution, antioxidant enzyme systems were formed in plants in order to scavenge ROS. SOD, CAT, and POD were important stress-resistant enzymes in the plant enzymatic antioxidant defense system [40]. The application of *T. asperellum* 22043 under salt stress decreased the MDA content in tomato and induced physiological protection of the plant against oxidative damage; similarly, the application of *T. harzianum* reduced the accumulation of MDA in salt-stressed wheat [41]. *T. asperellum* 22043 induced an increase in the activities of antioxidant enzymes such as POD, SOD, and CAT in salt-stressed tomatoes and enhanced the ability of tomato to scavenge reactive oxygen species [42]. Previous studies have found that *T. longibrachiatum* can increase the activity of resistance enzymes in wheat seedlings, thus improving the salinity tolerance of the plants [2]. In addition, previous studies have shown that *Trichoderma* can increase osmoregulatory substances content such as proline [43] and soluble protein to maintain the osmotic balance in plants under salt stress [19]. However, in the present study, *T. asperellum* 22043 caused a decrease in osmoregulatory substances content in salt-treated tomatoes. This may be due to the fact that *T. asperellum* 22043 increased the capacity of antioxidant and ion transporter in tomato. Tomato suffered lower salt damage and therefore did not need to accumulate osmoregulatory substances.

*Trichoderma* treatment usually upregulated many functional genes related to plant growth and stress tolerance at the transcriptome level in response to salt adversity [21]. Under salt stress, plants were severely poisoned by Na^+^ [44], and ion transport proteins can regulate Na^+^ homeostasis and enhance salt tolerance in plants. HKT was a key protein in the maintenance of Na^+^ homeostasis in plants [45]. The plasma-membrane Na^+^/H^+^ antiporter (SOS1) was capable of transporting excess Na^+^ out of the cell [46] and the vesicle-membrane Na^+^/H^+^ antiporter (NHX1) can store excess Na^+^ in the cytoplasm in its vesicles [47]. LHA4 was a plasma-membrane H^+^-ATPase that drives ATP hydrolysis and functions as a proton pump [48]. Previous studies have shown that overexpression of *AtNHX1-TsVP* significantly increased the tolerance of transgenic cotton to high salt stress [49]. In addition, upregulation of plasma-membrane Na^+^/H^+^ antiporter (SOS1) expression enhances salt tolerance in Arabidopsis [50]. Rubio [51] found that the expression of plasma-membrane Na^+^/H^+^ antiporter (SOS1) gene increased in tomatoes inoculated with *T. reesei* under salt stress conditions. The present results demonstrated that *T. asperellum* 22043 can regulate the transcriptional level of *HKT* gene so as to enhance salt tolerance in tomato by facilitating Na^+^ circulation. *T. asperellum* 22043 was able to regulate the expression of genes involved in oxidative stress to enhance the antioxidant defense systems in response to salt stress. A previous study has shown that *Trichoderma* induces upregulation of *CAT* and *SOD* gene expression in salt stress-treated cucumber roots to improve their antioxidant capacity [21]. Soil salinity was the result of the interaction of water and ionic stress, which adversely affects plant growth and development by disrupting the osmotic balance of plant cells. The main response of plants to salt stress was the inhibition of water uptake by the root system [52]. In most plants, water uptake and cross-cellular water flow in roots was mainly mediated by PIPs, which were the most abundant aquaporins in the plasma of plant cells and can participate in salt stress in plants [53]. *T. asperellum* 22043 were able to regulate *PIP2-9* gene expression to cope with salt stress. Similarly, a study found that aquaporin gene expression in tomato roots was reduced after inoculation with *T. harzianum* [54]. However, the specific role of individual genes or specific subfamilies in salt stress remains difficult to determine due to the complexity of aquaporins [55].

## 5. Conclusions

*T. asperellum* 22043 was able to mitigate the negative effects of salt stress on plants, including increasing the germination rate and shoot and root length of salt-stressed tomato seeds, as well as seedling survival, plant height, and chlorophyll content. Additionally, the increase in antioxidant enzymes activity was found to be very useful in providing tolerance to these plants under salt stress. Our results also indicated that *T. asperellum* 22043 was able to regulate the transcriptional level of the expression of the ion transport genes as well as the aquaporin genes to maintain the balance of Na^+^ as well as water in tomato cells under salt stress. Therefore, *T. asperellum* 22043 can be used as an environmentally friendly and effective strategy for saline soil to help plants combat salt stress through multiple mechanisms.

## Figures and Tables

**Figure 1 jof-11-00253-f001:**
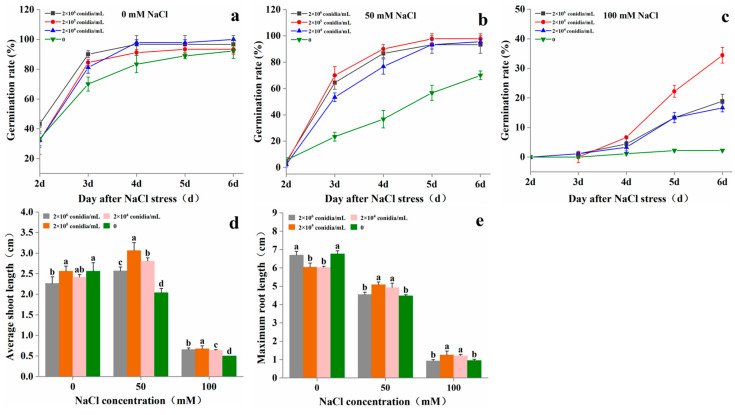
Effect of *T. asperellum* 22043 (0, 2 × 10^4^, 2 × 10^5^, 2 × 10^6^ conidia/mL spore suspensions) on the germination rates in tomato seeds under (**a**) 0, (**b**) 50, and (**c**) 100 mM NaCl stress and effect of *T. asperellum* 22043 (0, 2 × 10^4^, 2 × 10^5^, 2 × 10^6^ conidia/mL spore suspensions) on (**d**) the average shoot length and (**e**) the maximum root length in tomato seeds under 0, 50, and 100 mM NaCl stress. The line bars indicate the standard errors of the means (*n* = 3). Different letters in the graph represent significant differences according to one-way ANOVA between groups at *p* < 0.05.

**Figure 2 jof-11-00253-f002:**
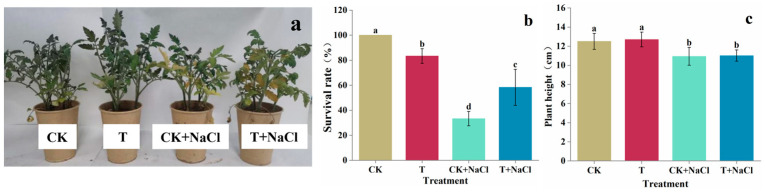
*T. asperellum* 22043 affects tomato seedlings’ growth. (**a**) shows the growth status of tomato plants, (**b**) their survival rate, and (**c**) the plant height after 14 days of NaCl treatment. CK indicates control tomato seedlings, CK + NaCl indicates tomato seedlings treated with 200 mM NaCl, T indicates tomato seedlings treated with *T. asperellum* 22043, and T + NaCl indicates tomato seedlings treated with *T. asperellum* 22043 under 200 mM NaCl stress. The line bars indicate the standard errors of the means (*n* = 3). Different letters in the graph represent significant differences according to one-way ANOVA between groups at *p* < 0.05.

**Figure 3 jof-11-00253-f003:**
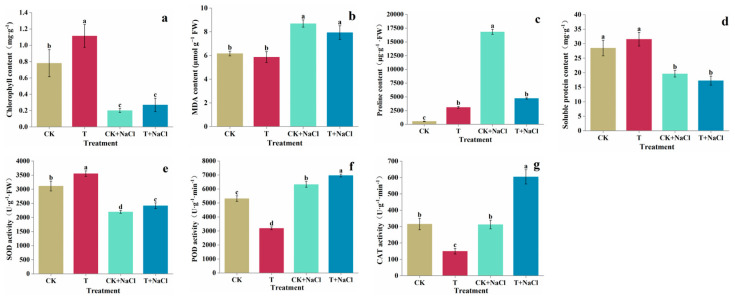
Effect on seedling (**a**) chlorophyll content, (**b**) MDA content, (**c**) proline content, (**d**) soluble protein content, (**e**) SOD activity, (**f**) POD activity, and (**g**) CAT activity under NaCl stress with and without *T. asperellum* 22043 in tomato seedlings. CK indicates control tomato seedlings, CK + NaCl indicates tomato seedlings treated with 200 mM NaCl, T indicates tomato seedlings treated with *T. asperellum* 22043, and T + NaCl indicates tomato seedlings treated with *T. asperellum* 22043 under 200 mM NaCl stress. The line bars indicate the standard errors of the means (*n* = 3). Different letters in the graph represent significant differences according to one-way ANOVA between groups at *p* < 0.05.

**Figure 4 jof-11-00253-f004:**
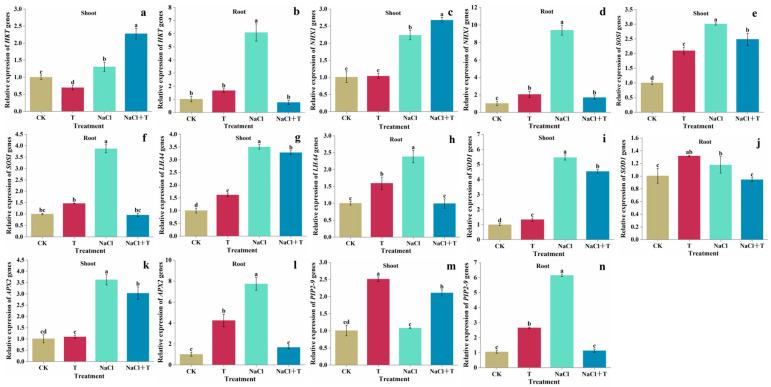
Effect of *T. asperellum* 22043 on the expression of (**a**,**b**) *HKT*, (**c**,**d**) *NHX1*, (**e**,**f**) *SOSI*, (**g**,**h**) *LHA4*, (**i**,**j**) *SOD1*, (**k**,**l**) *APX2*, and (**m**,**n**) *PIP2-9* genes in tomato seedling and root under salt stress. CK indicates control tomato seedlings, CK + NaCl indicates tomato seedlings treated with 200 mM NaCl, T indicates tomato seedlings treated with *T. asperellum* 22043, and T + NaCl indicates tomato seedlings treated with *T. asperellum* 22043 under 200 mM NaCl stress. The line bars indicate the standard errors of the means (*n* = 3). Different letters in the graph represent significant differences according to one-way ANOVA between groups at *p* < 0.05.

**Table 1 jof-11-00253-t001:** Sequences of the qRT–PCR primers used.

Gene Symbol	Forward Primer (5->3)	Reverse Primer (5->3)
*HKT*	TCTAGCCCAAGAAACTCAAAT	CTAATGTTACAACTCCAAGGAATT
*SOS1*	TCGAGTGATGATTCTGGTGG	ATCACAGTGTGGAAAGGCT
*NHX1*	CACGATATGGTGGGCTGGTT	GGGTGTGGCCAAATCTCGTA
*LHA4*	AAAGCAGAGAGAGAGAGACG	AGCACCACCCATTGAAAGGG
*SOD1*	AGCGGTGGTGTCTGTCTTAG	ACCCCAATTCAAAAGGCGTC
*APX2*	ATGGTAGCTGGAGGAGACC	TTGAGGGAGCATGGACCAAC
*PIP2-9*	CCTGGTTACAACAATGGAA	GGTCAGTAGCAGAGAAGA
*actin*	AAAAGTGCGAGTGTCCTGTCT	TCAAAAAAACAAATTGACTGG

## Data Availability

The original contributions presented in this study are included in the article. Further inquiries can be directed to the corresponding authors.

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
