# Peer review of "Trichoderma asperellum 22043: Inoculation Promotes Salt Tolerance of Tomato Seedlings Through Activating the Antioxidant System and Regulating Stress-Resistant Genes"

_jof, 2025, doi:10.3390/jof11040253_

Round 1
Reviewer 1 Report
Dear authors, your work is quite extensive and makes clear the role of T. asperellum 22043 in mediating the tolerance of tomatoes to saline soils. However, I have a few comments:
- In the title you use Trichoderma spp, when you are working with T. asperellum 22043. Please change.
- The summary is not clear, please restructure it
-
Change "2 μL of cDNA" in the qRT-PCR assays to the concentration used.
- In fungi, conidia/mL is used, not cfu/mL. Please change.
- I suggest that figure 2 be joined to graphs 3a and 3b.
- Improve the foot of figure 2, it is understood that it is a photograph.
- The section “3.5. Ion Transporter Gene Expression”. It seems more like a discussion than a presentation of results. Please restructure.
- I think the gene expression assays can be integrated, as the sections and descriptions are too short. I suggest generating a single figure with all the graphs and a single section.
-
I recommend reviewing the following papers to enrich the discussion:
“https://doi.org/10.1186/s40529-023-00368-x”
“https://doi.org/10.1007/s42729-019-00114-y”
https://doi.org/10.1016/j.catena.2018.12.009
https://doi.org/10.1007/978-3-030-91650-3_11
Figures 1d, 1e and 4 are not cited in the main text. Improve the arrangement of the figures so that they are more aesthetically pleasing.
Author Response
Dear Editors and Reviewer 1:
Thank you for your letter and for the reviewers’ comments concerning our manuscript entitled " Trichoderma spp. Inoculation Promotes Salt Tolerance of Tomato Seedlings Through Activating the Antioxidant System and Regulating Stress-resistant Genes” (ID: jof-3521247). We sincerely thank the reviewers for their constructive feedback and valuable suggestions. All comments have been carefully addressed, and modifications in the revised manuscript are highlighted in red text. Detailed point-by-point responses are provided below.
Comment 1: In the title you use Trichoderma spp., when you are working with T. asperellum 22043. Please change.
Response 1: Thank you for pointing this out. We agree with this comment. Therefore, we have changed Trichoderma spp. in the title to T. asperellum 22043 (Page 1, line 2).
Comment 2: The summary is not clear, please restructure it.
Response 2: We have rewritten the abstract, please see the revised Abstract (Page 1).
Comment 3: Change "2 μL of cDNA" in the qRT-PCR assays to the concentration used.
Response 3: Thank you for pointing this out. We agree with this comment. Therefore,
we have changed " 2 μL of cDNA " in the section 2.5 to " 20 ng of cDNA " (Page 4, line 161).
Comment 4: In fungi, conidia/mL is used, not cfu/mL. Please change.
Response 4: Thank you for pointing this out. We agree with this comment. Therefore, we have changed " cfu/mL " to " conidia/mL " throughout the article, including in the figures. (Page 3, line 105; Page 3, line 120; Page 5, lines 184-185, line 190; Page 5, figure 1 and caption, lines 196 and 198)
Comment 5: I suggest that figure 2 be joined to graphs 3a and 3b.
Response 5: Thank you for pointing this out. We agree with this comment. Therefore, we have connected figure 2 with figures 3a and 3b. (revised figure 2 and figure 3 at Page 6-7).
Comment 6: Improve the foot of figure 2, it is understood that it is a photograph.
Response 6: We have rewritten the title of figure 2, please see lines 212-214.
Comment 7: The section “3.5. Ion Transporter Gene Expression”. It seems more like a discussion than a presentation of results. Please restructure.
Response 7: Agree. We have recharacterized this section by removing the following references to discussion. The revised section you can find on pages 7, section 3.5, lines 247-255.
Sentence 1: T. asperellum 22043 was involved in the regulation of ion transporter genes to reduce Na+ accumulation in tomato;
Sentence 2: to limit the uptake of external Na+ by the roots;
Sentence 3: to facilitate Na+ circulation between seedlings and the root system;
Sentence 4: thus it was able to store the excess Na+ in the liquid vesicles;
Sentence 5:It was shown that under salt stress, tomato seedlings were able to excrete excess Na+ from the cells and reduce damage to other organelles in the tomato cytoplasm.
Comment 8: I think the gene expression assays can be integrated, as the sections and descriptions are too short. I suggest generating a single figure with all the graphs and a single section.
Response 8: Thank you for pointing this out. We agree with this comment. Therefore, we integrated figures 4, 5 and 6 from the gene expression section into a single figure and synthesized sections 3.5, 3.6 & 3.7 into a single section. We have changed the title of section 3.5 from “Ion Transporter Gene Expression” to “Gene Expression”. The revised figure is shown in figure 4 on page 8, and the revised section is section 3.5 on page 7.
Comment 9: I recommend reviewing the following papers to enrich the discussion:
“https://doi.org/10.1186/s40529-023-00368-x”
“https://doi.org/10.1007/s42729-019-00114-y”
https://doi.org/10.1016/j.catena.2018.12.009
https://doi.org/10.1007/978-3-030-91650-3_11
Response 9: Agree. We have added the above ref into different paragraphs in the Discussion. You can find them on lines 277, 280, 308, and 311, respectively.
Comment 10: Figures 1d, 1e and 4 are not cited in the main text. Improve the arrangement of the figures so that they are more aesthetically pleasing.
Response 10: We feel sorry for our carelessness. In our resubmitted manuscript, we have cited figures 1d and 1e on page 5, lines 188-189. We also have cited figures 4 on page 7, section 3.5. Figures 2, 3 and 4 are adjusted. The revised figures are shown on pages 6-8. Thanks for your correction.
We believe these revisions have significantly strengthened the manuscript. Thank you again for your constructive feedback. Please let us know if further modifications are needed.
Sincerely,
Hetong yang
On behalf of all authors
Reviewer 2 Report
“Trichoderma spp. Inoculation promotes salt tolerance of tomato seedlings through activating the antioxidant system and regulating stress-resistant genes”. This manuscript reports that T. asperellum 22043 helps mitigate the negative effects of salt stress on tomato plants. Furthermore, increased its antioxidant enzyme activity. The manuscript can be accepted for publication, with minor observations.
The quality of Figure 1 could be improved, especially the legends within the graphs—same case for the graphs in Figures 3 and 4.

Author Response
Dear Editors and Reviewer 2:
Thank you for your letter and for the reviewers’ comments concerning our manuscript entitled " Trichoderma spp. Inoculation Promotes Salt Tolerance of Tomato Seedlings Through Activating the Antioxidant System and Regulating Stress-resistant Genes” (ID: jof-3521247). We sincerely thank the reviewers for their constructive feedback and valuable suggestions. All comments have been carefully addressed, and modifications in the revised manuscript are highlighted in red text. Detailed point-by-point responses are provided below.
Comment 1: In section 2.1 Trichoderma strain and plant material, the authors need to added methodology including the morphological and/or molecular characterization of Trichoderma asperellum 22043.
Response 1: Agree. The morphology and DNA sequence (ITS and TEF1 genes) have been published in our previous study, Hu, J.; Chen, K.; Li, J.; Wei, Y.; Wang, Y.; Wu, Y.; ... & Denton, M. D. Large-scale Trichoderma diversity was associated with ecosystem, climate and geographic location. Environmental Microbiology 2020, 22(3), 1011-1024. https://doi.org/ 10.1111/1462-2920.14798. Therefore, we have added this reference into section 2.1. Please see lines 97 and 98.
Comment 2: The methodology must be more specific to allow replication of experiments, particularly analytical methods.
Response 2: Agree. We have added “The experimental design was 3 Trichoderma spore concentrations x 3 NaCl concentrations x 3 replicates.” and “The experimental design was 2 Trichoderma treatments (+ and -) x t salinity levels (+ and -) x 3 replicates.” into section 2.2 and 2.3, respectively. Please see lines 106 and 117.
Comment 3: Can you add a reference on how DMA content in plants can be used to evaluate the degree of exposure to salt stress?
Response 3: Agree. We have added a reference on line 294.
Comment 4: The quality of Figure 1 could be improved, especially the legends within the graphs—same case for the graphs in Figures 3 and 4.
Response 4: Thank you for this suggestion. We have modified the legend of Figure 1 and replaced the original chart with color chart to show more clearly the differences in the data (now figure 1, 2, 3, 4).
Comment 5: In Refences section, please describes how it is cited in the guide for authors: Journal Arti cles: Author 1, A.B.; Author 2, C.D. Title of the article. Abbreviated Journal Name Year, Volume, page range.
Response 5: We did follow the format as the reviewer suggested, such as Rubio, M. B.; Quijada, N. M.; Pérez, E.; Domínguez, S.; Monte, E.; Hermosa, R. Identifying Beneficial Qualities of Tricho-derma Parareesei for Plants. Appl Environ Microbiol 2014, 80 (6), 1864–1873. https://doi.org/10.1128/AEM.03375-13.
We believe these revisions have significantly strengthened the manuscript. Thank you again for your constructive feedback. Please let us know if further modifications are needed.
Sincerely,
Hetong yang
On behalf of all authors
Reviewer 3 Report
This paper provides details about the capabilities of, Trichoderma in alleviation of salt stresses in plants. Most of the information has been reported elsewhere but this paper provides details. Line 101. Suggest starting the sentence with 'Uniformly'.
Line 101. Suggest starting the sentence with 'Uniformly'.
Author Response
Dear Editors and Reviewer 3:
Thank you for your letter and for the reviewers’ comments concerning our manuscript entitled " Trichoderma spp. Inoculation Promotes Salt Tolerance of Tomato Seedlings Through Activating the Antioxidant System and Regulating Stress-resistant Genes” (ID: jof-3521247). We sincerely thank the reviewers for their constructive feedback and valuable suggestions. All comments have been carefully addressed, and modifications in the revised manuscript are highlighted in red text. Detailed point-by-point responses are provided below.
Comment 1: Line 101. Suggest starting the sentence with 'Uniformly'.
Response 1: Thank you for pointing this out. We agree with this comment. Therefore, we have modified “ Picked full, uniformly sized tomato seeds “to “Uniformly sized and full tomato seeds”. Please see line 101.
We believe these revisions have significantly strengthened the manuscript. Thank you again for your constructive feedback. Please let us know if further modifications are needed.
Sincerely,
Hetong yang
On behalf of all authors
Round 2
Reviewer 1 Report
Accepted
Accepted